How does serif vs sans serif typeface impact the usability of e-commerce websites?

Vecino Sara 1 uo264093@uniovi.es
Mehtali Jonas 2
de Andrés Javier 3
Gonzalez-Rodriguez Martin 1
http://orcid.org/0000-0002-5666-9809 Fernandez-Lanvin Daniel 1
1 Department of Computer Science, University of Oviedo , Oviedo, Asturias , Spain
2 UFR de Mathématique et d’Informatique, Université de Strasbourg , Strasbourg , France
3 Department of Accounting, University of Oviedo , Oviedo, Asturias , Spain
Asif Muhammad
Electronic publication date: 2022 Nov 18
Publication date: 2022
Volume: 8
Electronic Location ID: e1139
Received 2022 Aug 15; Accepted 2022 Oct 6
Copyright: © 2022 Vecino et al.
Copyright year: 2022
Copyright holder: Vecino et al.
License: This is an open access article distributed under the terms of the Creative Commons Attribution License, which permits unrestricted use, distribution, reproduction and adaptation in any medium and for any purpose provided that it is properly attributed. For attribution, the original author(s), title, publication source (PeerJ Computer Science) and either DOI or URL of the article must be cited.
License URL: https://creativecommons.org/licenses/by/4.0/

Keywords: Usability, Serif, Sans serif, E-commerce website, Typeface

Funding: Department of Science, Innovation, and Universities (Spain) National Program for Research, Development, and Innovation RTI2018-099235-B-I00 This work was funded by the Department of Science, Innovation, and Universities (Spain) under the National Program for Research, Development, and Innovation (project RTI2018-099235-B-I00). The funders had no role in study design, data collection and analysis, decision to publish, or preparation of the manuscript.

==============================
This study tries to find evidence that points towards the best typeface to use in e-commerce websites to maximize usability, trust, loyalty, appearance and overall user satisfaction. We tested the difference between serif and sans serif inside the same font family. A total of 246 volunteers participating in the experiment were asked to complete a set of tasks and a questionnaire on an e-commerce website prototype. We measured task completion time, reading speed and reading comprehension. From the results, using multiple linear regression, we deduced that only gender determines user preferences. Females tend to prefer the serif version of the typeface under study. Although most e-commerce websites use sans serif typefaces, we could not find evidence supporting this decision. The serif and sans serif characteristic inside the same font family does not affect usability on a website, as it was found that it has no impact on reading speed and user preference.

Introduction

The importance of our research relies on how useful it is for e-commerce websites to improve usability in order to gain clients. In the third quarter of 2021, users spent an average of 6 h and 58 min surfing the web (Kemp, 2022). Time spent online has steadily increased over the past years and users become increasingly more selective regarding their website activity (Statistics & Data, 2021). Online websites have to compete for the user’s attention in order to increase their traffic. Therefore there is a real economic incentive in attracting the user’s attention and thus, improving usability.

One of the parameters that affects readability, and thus usability on a website is the typeface that is used. It is unclear whether serif or sans serif is more desirable. Both types are preferred by users, compared to script and monospace, given their characteristics of font personality (Sasidharan & Dhanesh, 2008). There is a belief that serif typefaces improve readability, but some authors consider it a myth (Latin, 2017). In the case of the biggest e-commerce websites such as Amazon, Ebay or Walmart1 sans serif typefaces are preferred. Most important companies are also changing their logos to a simple sans serif text with their brand name, in contrast to the more complex logos they used to have (Sienkiewicz, 2020). We want to test whether this trend is supported by user preferences or performance on the website. Some studies evidence there is an difference in readability between serif and sans serif (Rello & Baeza-Yates, 2016; Dogusoy, Cicek & Cagiltay, 2016), while others state that the difference is only in user preference (Bernard et al., 2003; Boyarski et al., 1998).

Although previous works are interesting, they compare serif and sans serif typefaces of different families, which could be a bias given that it is not possible to know if the differences come from the serif/sans serif characteristic or from user preference towards one of the selected families.

In our study, we test how the difference between serif and sans serif typefaces of the same family affects usability, from an objective and subjective perspective. To obtain the data we performed two tests using a prototype of an e-commerce website, one that consists on completing some tasks and a reading test where participants have to read the description of a product of the website. For the reading test we measure speed as well as reading comprehension and warn about it before the test to avoid users skipping the task. To gather user preferences we use a questionnaire where usability, trust, loyalty and the appearance of the website are evaluated. We also added a specific typography question to assess the typeface preference.

The remainder of the article is structured in the following way. In the Background section the previous work is shown and commented. Then, in the Methods section, the development of the experiment and analysis of data are explained. The results are shown in the next section. In the Discussion we explain and interpret the results, ending with a Conclusion section.

Background

The impact of typeface on readability and usability has already been studied in several previous works and the debate between serif and sans serif has been going on for a long time.

Wallace et al. (2020) did a research with 63 participants where they studied font preference and effectiveness. Preference was evaluated with a toggle task where they tested a total of 16 fonts appearing in pairs. Effectiveness was evaluated with a reading speed and comprehension test where users read passages and answered questions about them. Each participant tested five fonts in the effectiveness test. Wallace et al. (2020) found that users read 51% faster in their fastest typeface. However, the most efficient typeface does not equal user preferences and 23% of their participants read the slowest in their preferred typeface.

Comparing serif and sans serif typefaces, Bernard et al. (2003) found that between Times New Roman and Arial there are no objective differences but there are in user perception, where Times New Roman, a serif typeface, was perceived by the participants to be more difficult to read.

Sasidharan & Dhanesh (2008) studied the differences in personality between serif, sans serif, monospace and script through a questionnaire that participants had to complete to measure trust after using a simulated banking website. Serif and sans serif were perceived as the ones that give the most trust, but no significant differences were found among them in the questionnaire results. Serif and sans serif typefaces were seen as business-like.

Boyarski et al. (1998) focused on reading comprehension and user preference between fonts designed for computer display and for print. One of the tests compares Georgia, a serif typeface designed for screen display, against Verdana, a sans serif typeface designed also for screens. In a direct comparison there was a slight preference for the sans serif typeface, but results were very similar.

Another study shows that what improves readability for people with dyslexia also benefits people without it (Rello & Baeza-Yates, 2016). They conclude that font type does have an impact on reading time, being sans serif and roman typefaces the ones that improve readability the most. There were 97 participants of which 48 have dyslexia.

Dogusoy, Cicek & Cagiltay (2016) conducted an eye tracking study with 10 participants where they compared how serif and sans serif influences reading on a screen focusing on misspellings. The compared typefaces were Times New Roman and Arial. They concluded that participants read faster and with more accuracy from the sans serif typeface. These results are limited due to small sample size.

Chatrangsan & Petrie (2019) did a study where they compared combinations of different font sizes, 14, 16 and 18 points, with serif and sans serif typefaces. They performed two experiments one in the UK and the other in Thailand, and separated the participants into two groups, older and younger. The typefaces used for the English experiment were Times New Roman as the serif typeface, and Arial for the sans serif one. Their study focused on reading on a tablet some texts, in English or Thai, depending of the country, and answering some questions about them. They measured reading time, ratings of ease and tiring of reading and preference for the combinations of typeface and font size. The results of the UK experiment were that typeface does not affect reading time but it does reading comprehension, as users answered correctly significantly more questions in the serif typeface. On the other hand, participants found the sans serif typeface easier to read, and older ones preferred the sans serif typeface. Chatrangsan & Petrie (2019) could not arrive to a conclusion of which font type is best for English text. In the Thai text experiment typefaces do not affect reading time but in combination with the font size, the 18 point sans serif one was found to be the fastest combination. As in the English experiment, serif performed better on reading comprehension and was also found to be easier to read and less tiring, as well as the preference of users. For Thai text, Chatrangsan & Petrie (2019) recommend the serif typeface.

Josephson (2008) compared the legibility of four serif and sans serif typefaces with an eye tracker. The serif typefaces were Times New Roman and Georgia. The sans serif typefaces were Arial and Verdana. The sample size was six participants and there were no statistically significant differences between the fonts, although they comment that differences found in descriptive statistics might indicate that statistically significant differences could be found if the study was repeated with a larger sample.

These studies compared the serif and sans serif characteristic with fonts with different styles such as Times New Roman vs Arial (Chatrangsan & Petrie, 2019; Bernard et al., 2003; Dogusoy, Cicek & Cagiltay, 2016). Our approach is to use typefaces that belong to the same font family so that no other factor is introduced in the comparison.

Other articles, such as the following ones, studied typeface and how to improve readability but they do not focus on the difference between the serif vs sans serif characteristic.

Pušnik, Možina & Podlesek (2016) studied how typeface, letter case and position on the screen affects the recognition of short words mainly in television broadcasts. In this study they selected five typefaces, Calibri, Trebuchet, Verdana, Georgia and Swiss 721, being the last one the only not designed for on-screen use. They did three sessions, one for each letter case, being these lower, upper and sentence case, and in each session they combined the five mentioned typefaces with four different positions on the screen having 20 sequences per session. There were 50 participants, 34 women and 16 men, and for each session one sequence was selected at random. Their final recommendation regarding typeface is to use typefaces with distinctive design characteristics such as Georgia, but it only applies to bold typefaces which are the ones they tested.

Pušnik, Podlesek & Možina (2016) studied how the recognition of letter improves while comparing lower case to upper case letters and also when increasing the x-height of the lower case letter to the main size of the upper case ones. They used five typefaces, Calibri, Georgia, Trebuchet, Verdana and Swiss 721, which are representatives of serif and sans serif typefaces. In this study bold typefaces were used, as it is common in television broadcasts. The experiment involved 50 participants, 25 females and 25 males. In the case of Georgia, Trebuchet and Verdana, results were similar when comparing lower case to upper case. For Calibri, the recognition improved when displayed in upper case. The worst performing typeface was in this case Swiss 721.

These two studies above mentioned (Pušnik, Možina & Podlesek, 2016; Pušnik, Podlesek & Možina, 2016), were designed for television broadcasts and used bold letter type. They are interesting as legibility is very important in the broadcasting field as well as in an e-commerce website, but the results can not be compared as the size of a computer screen and a television one differs, and the interaction of users is not the same with the two devices.

Grobelny & Michalski (2015) compared the influence that changing different factors of the design of a product has on people’s opinions about it. They changed the position of the brand name compared to the image, the background colour, and the font size of the brand name. A total of 60 students participated in the experiment. Although they did not compare typefaces or other factors regarding fonts, it is evidenced that typography can influence people’s opinion on commercial products, as there was a general preference towards big and compact captions.

Methods

Experiment development

We compared serif and sans serif characteristic as they are the ones that in Sasidharan & Dhanesh (2008) were found to give more trust to the user given the personality assigned to them, and we also performed the test on an e-commerce website that needs to be reliable.

To select the typefaces, we decided to choose the most popular web typefaces, given that our study is focused on website usability. We used the same font family only changing the serif and sans serif characteristic so that other factors of its appearance remained the same. The font ranking we used came from Google Fonts Analytics (Google, 2022) from the 14th of June of 2021 to the 14th of June of 2022. In this ranking, Roboto is the most popular typeface. It belongs to the sans serif category, so we chose its serif version, that is Roboto Serif to compare with. The font size used in our website is 16 px, the default size in Google Chrome, Firefox and Edge.

For the tests we created a prototype that is based on an e-commerce website for buying fruits and vegetables. It contains the main elements of e-commerce websites: A grid of products.

Product details pages.

A shopping cart.

Participants had to follow instructions to complete the experiment. They came in three phases: First task: Where users had to follow certain steps.

Second task: A reading test.

Questionnaire: Gathering anonymous user information and measuring usability and typeface preference.

The website was created so that the main page shows the instructions for the first task, that essentially asked the user to complete a full purchase. There, users are asked to find a specific product, add it to the cart and buy it. Once they click the continue button, users see the grid of products so that they can start with the task. When a user clicks on the buy button, they are redirected to the instructions of the second task. In this first task we wanted to measure the time it took to complete it, and how typeface influences the purchase process.

For the second task users had to read the details page of a specific product. In our case, the product was potatoes. The text was taken from a paragraph of the Potatoes page of Wikipedia. In this test we wanted to measure the reading speed and comprehension of the description of the product to obtain information about difference in typefaces while reading a longer text. Users were warned about the reading comprehension question in order to dissuade them from skipping directly to the questionnaire.

Last, the questionnaire consisted of gathering user information such as age and gender. It also included a reading comprehension question and a usability questionnaire. This will be further explained in a following section.

Screenshots of the website prototype we created can be seen in Figs. 1–5.

Figure 1 Grid of products on a mobile phone.

Figure 2 Details page of a product on a mobile phone.

Figure 3 Cart after a product was added on a mobile phone.

Figure 4 Part of the product description on a mobile phone.

Figure 5 Part of the questionnaire on a mobile phone.

The prototype was designed so that half of the users were shown a website with a serif typeface, and the other half with the sans serif one. It was implemented using HTML, CSS and JavaScript, and deployed on GitHub Pages. To implement the A/B tests we used Google Optimize, a tool that allowed us change the typeface for each of the tests and redirect the traffic of our website so that the same number of users would see each website version. The website has a responsive design and the data was gathered only for mobile phone users. We recruited the participants by sharing a link to our prototype through social media. We have three different languages in our website: English, Spanish and French, and was distributed mainly in Spain, France, Pakistan and Lithuania.

The objective metrics we gathered from the experiment were: Time to complete the tasks.

Time to read the description (reading speed).

Correct answer to the question about the product description (reading comprehension).

Questionnaire

The last part of the experiment was a questionnaire that assessed the user’s experience. We compared the objective measurements to the subjective ones obtained by this method. The questionnaire is based on the Standardized User Experience Percentile Rank Questionnaire (SUPR-Q) (Sauro, 2015), a study that analyzed over 4,000 responses that users gave after using more than 100 websites in order to generate a questionnaire to evaluate the quality of a website. This questionnaire has eight items and contains four factors: usability, trust, appearance and loyalty. To these items we added one question related to typography, item 9, taken from Faisal et al. (2017). In Faisal et al. (2017) they studied the user preferences for a series of web design attributes which include typography among others. As a part of their results and analysis, Faisal et al. (2017) concluded that typography influenced trust and satisfaction in a positive way, as font color, proper line spacing and typeface with a readable font size results in loyalty because of the satisfactory and trustworthy appearance. This is a list of the items that we used to measure the users opinions on the website: The website is easy to use. (usability)

It is easy to navigate within the website. (usability)

I feel comfortable purchasing from the website. (trust)

I feel confident conducting business on the website. (trust)

How likely are you to recommend this website to a friend or colleague? (loyalty)

I will likely return to the website in the future. (loyalty)

I find the website to be attractive. (appearance)

The website has a clean and simple presentation. (appearance)

It is easy to read the text on this website with the used font type. (typography)

As in Sauro (2015), all the questions were to be answered using a five-point Likert scale, except for the question “How likely are you to recommend this website to a friend or colleague?” which has a 11-point scale.

The SUPR-Q score is calculated in the following way:

(1) score=(sumofpointsfromresponseswithoutquestions5and9/8)+(responseofquestion5/2)

Item 9 of the questionnaire is not added to the score, but used on its own to obtain information about user preferences for each typeface. This item also used a five-point Likert scale as it is the number used in Sauro (2015).

The complete questionnaire that gathers the user’s personal information, the usability information and the reading comprehension question consists on the following items: Birthdate.

Gender.

How much time are potatoes boiled in order to become soft? Between 20 and 30 min.

Between 10 and 25 min.

Between 20 and 25 min.

The website is easy to use.

It is easy to navigate within the website.

I feel comfortable purchasing from the website.

I feel confident conducting business on the website.

How likely are you to recommend this website to a friend or colleague?

I will likely return to the website in the future.

I find the website to be attractive.

The website has a clean and simple presentation.

It is easy to read the text on this website with the used font type.

For the birthdate, the user chose numbers from three different combo boxes, for day, month and year. In the case of the gender item, the user chose from a combo box with the options: female, male and other. The third item is the reading comprehension question, to answer it, users had already read in the previous page a snippet of a Wikipedia article and then they were presented with this question to assess if they have read it carefully or not. Users were warned in the instructions before the reading task that they had to answer a question about the paragraph.

Data analysis

Pre-processing

Here are the different pre-processing filters we applied to the raw data: Filtering invalid data: We removed the duplicates in our data, as well as invalid ones, such as birth dates from 2019 on, or users that did not get one of the two typefaces due to issues with their navigator blocking Google Optimize.

Description time normalization: We normalized the time it took the users to read the description of the product, that is the reading task, as we have users reading in French, English and Spanish. We did it by multiplying the user’s reading time by the number of characters of their text over the number of characters in the English version.

(2) normalizedtime=numberofreadcharactersnumberofcharactersinEnglish

Winsorizing: In the data we gathered, we found some time values abnormally large, and that can be explained by users leaving their phone for some minutes and then continued doing the tests. Those values are invalid as they do not reflect the normal use of a website and affect the time. We decided to substitute all the values above the 95 percentile by the value of the 95 percentile.

Processing

To process the data we used regression analysis. In particular, we applied multiple linear regression. We applied this method because our sample size exceeds the general rule of 10 observations per parameter to be estimated, but it was not enough to use more complex methods such as for example quantile regression or non-parametric models (see, among others, Greene (2018)). This is the general equation used:

(3) predicted_variable=const+(typeface_coef)∗x1+(age_coef)∗x2+(gender_coef)∗x3+ϵ

These are the dependent variables: Reading comprehension.

Total time.

Reading task time.

SUPR-Q score.

Typography score.

Typography score for Roboto.

Typography score for Roboto Serif.

These are the independent variables: Typeface: x1 in the equation.

Age: x2 in the equation.

Gender: x3 in the equation.

The typeface is the main variable of the study, being age and gender control variables.

With the results we can deduce if the typeface, age or gender of the participants have an impact on each of the variables. In addition to this, we also gathered basic statistics of the time it took the users to complete the tasks and the questionnaire.

Both processing and pre-processing were done using Python libraries: Pandas, Scipy and Scikit-learn.

Results

Out of 246 participants, 134 were males and 112 females, with a mean age of 27 years. A total of 179 users completed the experiment in English, 50 in Spanish and 17 in French. A total of 83 failed to answer correctly to the reading comprehension question. This leaves us with 163 that answered correctly which is 66%.

Table 1 shows some descriptive features of the Roboto and Roboto Serif subsamples. Table 2 represents basic statistics of the time spent by users on the first task, the second task and the questionnaire measured in seconds.

Table 1 Results of metrics for Roboto and Roboto Serif.

	Roboto	Roboto Serif	
Total number of participants	110	136	
Mean typography score	7.84	7.93	
Mean female typography score	7.963	8.414	
Mean male typography score	7.714	7.564	
Mean SUPR-Q score	7.12	7.16	
Percentage of reading comprehension correct answers	65%	68%	

Table 2 Basic statistics for the time of the tasks and questionnaire.

	First task	Second task	Questionnaire	
Typeface	Mean	Standard deviation	Mean	Standard deviation	Mean	Standard deviation	
Roboto	27.892	14.560	62.103	35.025	88.357	29.899	
Roboto Serif	30.228	18.596	58.176	36.329	84.670	26.979	

Multiple linear regression analysis

These are the results of the regression analysis along with their specific equation:

Reading comprehension

(4) total_time=0.769+0.022∗x1−0.003∗x2−0.100∗x3

Total time

(5) total_time=210300−7053.417∗x1+239.459∗x2−11180∗x3

Reading test time

(6) reading_test_time=62960−3992.632∗x1−16.353∗x2−843.493∗x3

SUPR-Q score

(7) SUPR−Q_score=7.362+0.055∗x1−0.017∗x2+0.452∗x3

Typography score

(8) typography_score=8.334+0.105∗x1−0.029∗x2+0.579∗x3

Typography score for Roboto

(9) typography_score_roboto=8.862−0.043∗x2+0.298∗x3

Typography score for Roboto Serif

(10) typography_score_robotoserif=8.028−0.017∗x2+0.835∗x3

Discussion

In Tables 3–9, the results of the multiple linear regression analysis are stated.

Table 3 Regression analysis results for the predicted variable reading comprehension.

Constant	Typeface	
Parameter estimation	p-value	t-statistic	Parameter estimation	p-value	t-statistic	
0.769	0.000	7.955	0.022	0.712	0.369	
Age	Gender	
Parameter estimation	p-value	t-statistic	Parameter estimation	p-value	t-statistic	
−0.003	0.355	−0.927	−0.100	0.100	−1.651	
Adj R2	F-statistic (p-value)	
0.003	1.284 (0.281)	

Table 4 Regression analysis results for the predicted variable total time.

Constant	Typeface	
Parameter estimation	p-value	t-statistic	Parameter estimation	p-value	t-statistic	
210,300	0.000	13.974	−7,053.417	0.457	−0.745	
Age	Gender	
Parameter estimation	p-value	t-statistic	Parameter estimation	p-value	t-statistic	
239.459	0.604	0.519	−11,180	0.238	−1.183	
Adj R2	F-statistic (p-value)	
−0.004	0.715 (0.544)	

Table 5 Regression analysis results for the predicted variable reading time.

Constant	Typeface	
Parameter estimation	p-value	t-statistic	Parameter estimation	p-value	t-statistic	
62,960	0.000	8.581	−3,992.632	0.388	−0.865	
Age	Gender	
Parameter estimation	p-value	t-statistic	Parameter estimation	p-value	t-statistic	
−16.353	0.942	−0.073	−843.493	0.855	−0.183	
Adj R2	F-statistic (p-value)	
−0.009	0.255 (0.857)	

Table 6 Regression analysis results for the predicted variable SUPR-Q score.

Constant	Typeface	
Parameter estimation	p-value	t-statistic	Parameter estimation	p-value	t-statistic	
7.362	0.000	16.229	0.055	0.847	0.193	
Age	Gender	
Parameter estimation	p-value	t-statistic	Parameter estimation	p-value	t-statistic	
−0.017	0.219	−1.233	0.452	0.114	1.587	
Adj R2	F-statistic (p-value)	
0.004	1.351 (0.259)	

Table 7 Regression analysis results for the predicted variable typography score.

Constant	Typeface	
Parameter estimation	p-value	t-statistic	Parameter estimation	p-value	t-statistic	
8.334	0.000	17.993	0.105	0.719	0.361	
Age	Gender	
Parameter estimation	p-value	t-statistic	Parameter estimation	p-value	t-statistic	
−0.029	0.043	−2.034	0.579	0.048	1.990	
Adj R2	F-statistic (p-value)	
0.021	2.728 (0.045)	

Table 8 Regression analysis results for the predicted variable typography score for Roboto.

Constant	Age	
Parameter estimation	p-value	t-statistic	Parameter estimation	p-value	t-statistic	
8.862	0.000	12.586	−0.043	0.069	−1.835	
Gender		
Parameter estimation	p-value	t-statistic		
0.298	0.521	0.643				
Adj R2	F-statistic (p-value)	
0.015	1.828 (0.166)	

Table 9 Regression analysis results for the predicted variable typography score for Roboto Serif.

Constant	Age	
Parameter estimation	p-value	t-statistic	Parameter estimation	p-value	t-statistic	
8.028	0.000	15.259	−0.017	0.323	−0.992	
Gender		
Parameter estimation	p-value	t-statistic		
0.835	0.026	2.251		
Adj R2	F-statistic (p-value)	
0.031	3.124 (0.047)	

After conducting the analysis with the reading comprehension as the predicted variable we found that the typeface, age and gender have no significant impact on this variable, according to the p-values in Table 3.

In Tables 4–6, given that the p-values are all higher than 0.05, we can deduce that neither typeface, age or gender affect the total time of the experiment, the completion time for the second task, the reading one or the SUPR-Q score, that measures the usability of the website. For the typography score, in Table 7, the only p-values lower than 0.05 are those of the age and gender, meaning these variables might have an impact on that score. To know more in detail how those characteristics affect the score, we performed regression analysis in the typography score for both Roboto and Roboto Serif.

For Roboto, the sans serif typeface, as seen in Table 8, the p-value for age is 0.068, between 0.05 and 0.1, which could indicate that age has a slight impact on the typography score, but given that the p-value of the F-statistic is higher than 0.05, we cannot say that age affects user preferences.

For Roboto Serif, the serif typeface, gender makes a difference in the typography score according to the p-value of gender being less than 0.05 in Table 9 and the p-value of the F-statistic being also lower than 0.05. Females tend to prefer this typeface compared to males.

Conclusion

The first conclusion obtained from the analysis is that there are no significant differences in user typeface preference and usability between Roboto and Roboto Serif. There are also no significant differences in reading comprehension of texts written in serif or sans serif in the same font family, and the same is true for task completion time. Regarding gender, the serif typeface is more preferred by female participants than male ones but no differences were found for the sans serif one.

With these statements we can conclude that the serif or sans serif characteristics inside of the same font family do not impact web usability. Thus our results join the ones found in Chatrangsan & Petrie (2019) that show that typeface does not affect reading time, but it contradicts the results of Rello & Baeza-Yates (2016) and Dogusoy, Cicek & Cagiltay (2016). These conclusions are important for the industry given that most e-commerce websites are currently using sans serif typefaces, thus it could be studied whether different styles of typefaces do affect usability and readability.

Though, these results might be different if tested with typefaces of different styles, given that our study is limited to only one font family. Our research is also limited due to having performed our experiments in an uncontrolled environment.

Our results could be used along with an automatic system for user classification. Steps have been taken in this direction as it can be seen in the work of Fernandez-Lanvin et al. (2018), that focused on the customization of e-commerce websites given automatic profiles of the users. It could be implemented in a way that when the system detects that the user is a female, a serif typeface is shown as opposed to the sans serif typefaces that are more popular on the web. On a website without an automatic system that changes the style, we encourage designers and developers to use both types of fonts as we found that inside the same font family they do not impact usability.

For future work we will conduct experiments in a controlled environment using an eye-tracker to gather more measurements relative to readability such as fixations on call-to-action buttons, number of fixations and fixation duration on the text of the website. It would be interesting as well to measure other characteristics of typefaces and website layout such as font size and line length. We will also consider the introduction of variables that measure different aspects of culture into the analysis as many e-commerce websites can expect to receive users from different countries. To do so, we will use some works that propose cultural dimensions and scores for countries, such as those by Hofstede (see, e.g., Hofstede (2001)).

Supplemental Information

Supplemental Information 1 Dataset containing all the information gathered from the participants.

The answers to the questionnaire: the time each user was on each page, the user agent they used, and mouse information.

Click here for additional data file.

Supplemental Information 2 Questionnaire of the experiment.

Click here for additional data file.

We would like to thank all the participants for their time in completing the experiment.

Additional Information and Declarations

Competing Interests

Author Contributions

Data Availability

1 These websites were last accessed in July 2022.

The authors declare that they have no competing interests.

Sara Vecino conceived and designed the experiments, performed the experiments, analyzed the data, performed the computation work, prepared figures and/or tables, authored or reviewed drafts of the article, and approved the final draft.

Jonas Mehtali conceived and designed the experiments, performed the experiments, analyzed the data, performed the computation work, prepared figures and/or tables, authored or reviewed drafts of the article, and approved the final draft.

Javier de Andrés conceived and designed the experiments, authored or reviewed drafts of the article, and approved the final draft.

Martin Gonzalez-Rodriguez conceived and designed the experiments, authored or reviewed drafts of the article, and approved the final draft.

Daniel Fernandez-Lanvin conceived and designed the experiments, authored or reviewed drafts of the article, and approved the final draft.

The following information was supplied regarding data availability:

The raw data are available in the Supplemental File.

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
