# Peer review of "How does serif vs sans serif typeface impact the usability of e-commerce websites?"

_PeerJ Computer Science, doi:10.7717/peerj-cs.1139_

## Round 0.1 · original submission · Minor Revisions

Dear authors, please improve the paper according to the experts' suggestions and resubmit. Thank you.

Reviewer 1 ·

Basic reporting

This is an interesting paper to test the usability about the used of fonts in e-commerce website. The structure and design is good and I provide few points for authors to improve the paper.
1. I suggest the author can provide more detail about the theoristics background of the paper. Especially Standardized User Experience Percentile Rank Questionnaire (SUPR-Q) Sauro (2015), and Faisal et al. (2017). It is not really clear in current description.

Experimental design

1. I suggest the authors should provide full questionnaire that used in the paper, which is important for reader to understand the results.

Validity of the findings

no comment

Additional comments

no comment

Reviewer 2 ·

Basic reporting

Regarding the communication, the manuscript is not well-structured; it is suggested to make edits to the terms and structure of the text body, e.g., line no "In Q3 2021". I hope it will help you to clarify the manuscript.

Experimental design

The experimentation and data gathering section requires more details on how the participants were recruited in this study and how the guidelines related to participation and experimentation were provided. In the results section, most results are presented clearly. However, the used terms, e.g., param. est., std dev in the table requires full words; otherwise, place them as the full name as notes below the table. It is also advised to add more details on the employed statistical approaches and how these tests were appropriate for the employed measures and samples. Further analysis may also be useful by exploring cultural perspectives.

Validity of the findings

The discussion is limited due to the missing explanation of how designers and developers should incorporate the study's results into an actual user interface.

Additional comments

The study's objective is to determine the impact of topographical aspects such as typeface on usability and other motives such as trust, loyalty, and satisfaction. The authors use subjective and objective approaches to explore the proposed measures of the study. The study is interesting to look inside of the design aspects to improve the usability of the websites or other affective motives. Overall, the paper contains sufficient information to justify the study's contribution. The authors swiftly make the audience understand the importance of the research that they are doing. The study presents results in those typographical aspects such as typeface on website usability and related aspects. The findings are somehow interesting because literature concerning the role of Topographical aspects in the e-commerce environment is scarce.
This paper presents a good review of e-commerce usability and typeface background. However, more material could be added regarding typeface and other measures such as trust, satisfaction, and loyalty. So, it would be advisable to add more evidence related to the employed aspects in separate subsections "typeface and (trust, loyalty, satisfaction, and usability measures). Some other important studies (e.g. (1) Pušnik, Nace, Anja Podlesek, and Klementina Možina. "Typeface comparison− Does the x-height of lower-case letters increased to the size of upper-case letters speed up recognition?" International Journal of Industrial Ergonomics 54 (2016): 164-169. (2) Pušnik, Nace, Klementina Možina, and Anja Podlesek. "Effect of typeface, letter case, and position on recognition of short words presented on-screen." Behaviour & Information Technology 35.6 (2016): 442-451". "Grobelny, Jerzy, and Rafał Michalski. "The role of background color, interletter spacing, and font size on preferences in the digital presentation of a product." Computers in Human Behavior 43 (2015): 85-100." That includes important approaches and observations related to the appropriateness of typeface and its impact on legibility and recognition should also be included in the in-literature review section and the results section. I would suggest the authors compare how the result from this work is different from the abovementioned studies. Accordingly, the authors should include a more valuable theoretical gap by reviewing the existing literature and conducting an in-depth analysis.

---

## Round 0.2 · accepted · Accept

Thank you for your fine contribution.

Reviewer 1 ·

Basic reporting

The authors have responsed to my comment and revised the paper.

Experimental design

N/a

Validity of the findings

N/a

Additional comments

n/a

Reviewer 2 ·

Basic reporting

The authors have accommodated all the changes. My recommendation is to accept it

Experimental design

ok

Validity of the findings

ok